# Laplace Approximation Based Epistemic Uncertainty Estimation in 3D Object Detection

**Peng Yun** [*]
Department of Computer Science and Engineering
Hong Kong University of Science and Technology
`pyun@cse.ust.hk`

**Ming Liu** [†]
Hong Kong University of Science and Technology
(Guangzhou)
`eelium@ust.hk`

**Abstract:** Understanding the uncertainty of predictions is a desirable feature for perceptual modules in critical robotic applications. 3D object detectors are neural networks with high-dimensional output space. It suffers from poor calibration in classification and lacks reliable uncertainty estimation in regression. To provide a reliable epistemic uncertainty estimation, we tailor Laplace approximation for 3D object detectors, and propose an Uncertainty Separation and Aggregation pipeline for Bayesian inference. The proposed Laplace-approximation approach can easily convert a deterministic 3D object detector into a Bayesian neural network capable of estimating epistemic uncertainty. The experiment results on the KITTI dataset empirically validate the effectiveness of our proposed methods, and demonstrate that Laplace approximation performs better uncertainty quality than Monte-Carlo Dropout, DeepEnsembles, and deterministic models.

**Keywords:** Laplace approximation, epistemic uncertainty, 3D object detection

## 1 Introduction

Detecting key objects in 3D space is an important task in robotic applications. It provides both semantic and spatial information for decision-making. Due to the success of deep learning, existing detectors have got great performance on benchmarks [1–3]. However, robots directly interact with the real world, the data distribution of which may drift over time and space. It is essential for a robotic perceptual module to know when it is uncertain, since the consequences of mistakes can be fatal in critical applications, like autonomous driving.

Laplace approximation (LA) [4] provides principled uncertainty estimation as other Bayesian approaches [5–10], and computes posterior weight distributions in terms of local curvatures. It can convert a deterministic model, trained under the maximum-a-posteriori framework, into a Bayesian model for uncertainty estimation, which is in a post-hoc way without changing the training procedure. Recent research reduces the computational costs on computing local curvatures with various approximations [11–15], and makes Laplace approximation applicable in modern neural networks. However, existing research is limited to end-to-end networks, and rare has been applied to networks with high-dimensional output space, like 3D object detection.

Neural networks with high-dimensional output space require Monte Carlo-based Bayesian inference (MCBI) to approximate the predictive distribution. The MCBI is not applicable in two-stage 3D detectors due to its non-end-to-end architecture. A two-stage detector contains two parts: a Region Proposal Network (RPN) generates 3D proposals, and an RCNN selects and refines a subset of proposals $S_p$ to get final results. The different weight samples in MCBI have different selections $S_p$ which are mis-matched in order as shown in Figure 1 (a). It causes overestimating the epistemic uncertainty in predictive distributions, as shown in Figure 1 (c). We propose an Uncertainty Separation and Aggregation (U-SPA) method to apply Laplace approximation in two-stage detectors, as shown in Figure 1 (b). It separates the uncertainty estimation in RPN and RCNN and aggregates them in the final predictive distribution (d).

---

[*]Peng Yun is also with Clear Water Bay Institute of Autonomous Driving, Nanshan, Shenzhen.

[†]Ming Liu (corresponding author) is also with Hong Kong University of Science and Technology, Hong Kong SAR, China and HKUST Shenzhen-HongKong Collaborative Innovation Research Institute, Futian, Shenzhen.

6th Conference on Robot Learning (CoRL 2022), Auckland, New Zealand.

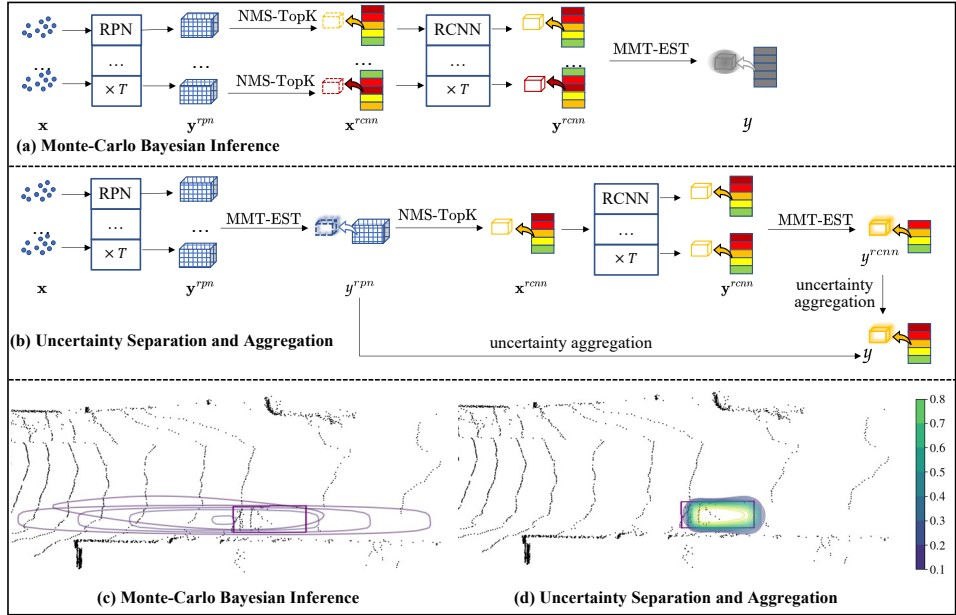

Figure 1: Bayesian inference pipeline for two-stage 3D detectors. (a) Monte-Carlo Bayesian Inference pipeline and (c) bird's-eye-view visualization of predictive distribution $y$. (b) Uncertainty Separation and Aggregation pipeline and (d) visualization of $y$. The contours reflect the objectiveness probability. The purple box denotes the ground truth.

In this paper, we tailor Laplace approximation methods to estimate epistemic uncertainty for 3D object detection under the context of autonomous driving. (1) We propose an Uncertainty Separation and Aggregation (U-SPA) pipeline to solve the mismatching problem in Monte-Carlo-based Bayesian inference of two-stage 3D object detectors. (2) We apply Laplace approximation to state-of-the-art 3D object detection networks, including both one-stage and two-stage detectors, and also provide a benchmark on the KITTI dataset to compare various Laplace approximation approaches, including LA (diagonal Fisher), LA (empirical diagonal Fisher), and sub-net LA. The proposed Laplace-approximation method can easily convert off-the-shelf 3D object detectors into a Bayesian neural network. The experiment results show that the Laplace approximation performs consistently better uncertainty quality than deterministic, MCDropout, and DeepEnsembles baselines.

## 2 Related Works

### 2.1 Laplace approximation

The major challenge in applying Laplace approximation [4] to large-scale neural networks is its expensive computation on the Hessian matrix. Even though the Fisher information matrix can be its equivalent, the exact Fisher requires storing and manipulating an $O(P^2)$ matrix, which is impractical to deploy in modern neural networks with the limited computational resources on mobile robots. Diagonal Fisher approximation is the lightest weight approach to approximate the Hessian. It ignores the correlation among model parameters and largely reduces the Fisher matrix size into $O(P)$, which has been well applied to second-order optimization [16] as well as incremental learning [17]. To balance the curvature information and computational efficiency, some researchers approximate the exact Fisher matrix with Kronecker Factorized Fisher [15] and low-rank approximations [14]. There also exists research applying Laplace approximation to a subset of model parameters [11–13], called SubNet-LA. Besides, even though empirical Fishers lack a theoretical foundation [18], it was adopted low-cost surrogate of Fisher information matrix and got success in practical applications [19–21]. This paper considers the diagonal Fisher approximation, empirical Fisher implementation, and subnet Laplace approximations in 3D object detection.

### 2.2 LiDAR-based 3D object detection

The geometrical information in LiDAR scans can be used to classify and locate key objects in 3D space. 3D object detector methods can be categorized into one-stage and two-stage methods. One-stage 3D detectors are end-to-end networks which consume LiDAR scans and estimate 2D

Figure 2: Illustration of one-stage (a) and two-stage 3D detectors (b). The one-stage detector also acts as an RPN subnetwork in two-stage detectors. (c) shows the RCNN subnetwork architecture. The "MLP" is short for Multi-layer perceptrons.

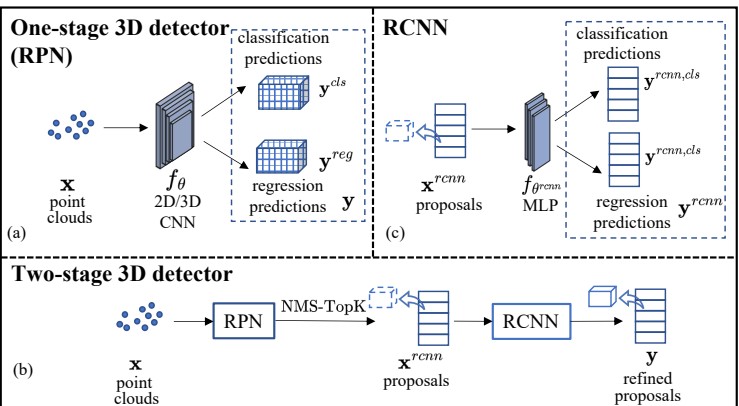

maps or 3D volumes for classification and regression [22–25], as shown in Figure 2 (a). Its output $\mathbf{y} = \{\mathbf{y}^{cls}, \mathbf{y}^{reg}\}$ contains classification predictions $\mathbf{y}^{cls}$ and regression predictions $\mathbf{y}^{reg}$. Yan et al. [24] exploited the sparsity of LiDAR scans and replaced the conventional dense convolution layers with sparse convolution operations. Lang et al. [25] extracted features from point clouds with PointNet [26] estimate classification and regression maps on bird's-eye-view images.

Two-stage detectors [27–29] contains two subnetworks: Region Proposal Network (RPN) and RCNN, as shown in Figure 2 (b). RPN generates proposals from point clouds with the same structure as one-stage detectors. A non-maximization-suppression (NMS) module then selects top-K unoverlapped proposals in terms of their classification score from RPN proposals $\mathbf{y}^{rpn}$. RCNN network refines these proposals with multiple linear layers, as shown in Figure 2 (c). The two-stage pipeline was proposed in [27], where Shi et al. adopted a PointNet-based RPN and generated point-wise proposals, and further refined these proposals with an MLP-based RCNN sub-network. To balance accuracy and computational efficiency, their follow-up work [28, 30] adopt sparse-convolution-based RPN to generate proposals in 3D volume. In this paper, we adopt one-stage 3D detectors, PointPillar [25] Second [24], and two-stage detectors PointRCNN [27] to demonstrate the effectiveness of Laplace approximation in 3D object detection.

### 2.3 Uncertainty estimation in object detection

Some recent works estimate uncertainties in robotic perceptual modules. Hall et al. [31] extended object detection to probabilistic object detection and quantified spatial and semantic uncertainties of detections with Monte-Carlo Dropout. The idea of probabilistic object detection was further extended to 3D object detection [32–34]. Feng et al. [32, 33] made an one-stage detector PIXOR [23] to estimate aleatoric uncertainties by adding an additional regression layer, as [35]. The estimated uncertainties were evaluated independently for each output dimension with Expected Calibration Errors (ECE) [36, 37]. To evaluate the general predictive distribution for LiDAR-based 3D object detection, Wang et al. [34] proposed mAP(JIoU), which evaluates the differences between the estimated predictive distribution and a ground-truth predictive distribution. They proposed a generative model to obtain the ground-truth predictive distribution. In our work, we evaluate the uncertainty quality with ECE following the method in[32], and also report the negative log-likelihood (NLL) performance as well as mAP(JIoU) [34].

## 3 Methodology

### 3.1 Preliminaries

Given a dataset $\mathcal{D} = \{\mathbf{X}, \mathbf{Y}\} = \{(\mathbf{x}_i, \mathbf{y}_i) | i = 1, ..., N\}$, we have a neural network $f_\theta(\mathbf{x})$ with its prior weight distribution $p(\theta)$. We want to derive the posterior weight distribution $p(\theta|\mathbf{X}, \mathbf{Y})$, so that we can conduct Bayesian inference and get the predictive distribution with

$$p(\mathbf{y}^*|\mathbf{x}^*, \mathbf{X}, \mathbf{Y}) = \int p(\mathbf{y}^*|\mathbf{x}^*, \theta)p(\theta|\mathbf{X}, \mathbf{Y})d\theta, \tag{1}$$

where $p(\mathbf{y}^*|\mathbf{x}^*, \theta)$ depicts the predictive model. In classification tasks, it is commonly defined as a categorical distribution with $p(\mathbf{y}^c = 1|\mathbf{x}, \theta) = \text{softmax}^c(f_\theta(\mathbf{x}))$, where the superscript $c$ denotes the $c-$th element of a vector. In regression tasks, it can be defined as $p(\mathbf{y}|\mathbf{x}, \theta) = \mathcal{N}(\mathbf{y}; f_\theta(\mathbf{x}), \tau^{-1}\mathbf{I})$. The $\tau$ can be a hyper-parameter or an estimation from data. This Bayesian inference equation catches the epistemic uncertainty caused by the weight distribution $p(\theta|\mathbf{X}, \mathbf{Y})$.

| **Algorithm 1:** Monte-Carlo Bayesian Inference (MCBI) | **Algorithm 2:** Uncertainty Separation and Aggregation (U-SPA) |
|---|---|
| **Input** : $\mathbf{x}$: point cloud ; $p(\theta\|\mathbf{X},\mathbf{Y})$: posterior weight distribution ; $T$: number of MC samples 
 **Output** : $y$: a set of 3D bounding-box predictive distributions 

 $preds \leftarrow \{\}$ ; 
 **for** $i \leftarrow 1$ **to** $T$ **do** 
 $\quad \theta \leftarrow$ sample from $p(\theta\|\mathbf{X},\mathbf{Y})$ ; 
 $\quad \mathbf{y} = f(\theta,\mathbf{x})$ ; 
 $\quad$ append $\mathbf{y}$ into $preds$ ; 
 **end** 
 $y \leftarrow$ MMT-EST$(preds)$; 
 return $y$ ; | **Input** : $\mathbf{x}$: point cloud ; $p(\theta\|\mathbf{X},\mathbf{Y})$: posterior weight distribution ; $T$: number of MC samples 
 **Output** : $y$: a set of 3D bounding-box predictive distributions 

 $y^{rpn} \leftarrow$ MCBI$(\mathbf{x}, p(\theta_{rpn}\|\mathbf{X},\mathbf{Y}), T)$ ; 
 /* $\overline{y^{rpn}}$ is the mode of $y^{rpn}$ */ 
 $\mathbf{y}^{rpn} \leftarrow \overline{y^{rpn}}$; 
 $S_p \leftarrow$ NMS-TopK$(\mathbf{y}^{rpn})$ ; 
 $\mathbf{x}^{rcnn} \leftarrow \{\mathbf{y}_i^{rpn}\|i \in S_p\}$ ; 
 $y^{rcnn} \leftarrow$ MCBI$(\mathbf{x}^{rcnn}, p(\theta_{rcnn}\|\mathbf{X},\mathbf{Y}), M)$ ; 
 $y \leftarrow$ aggregate $y^{rpn}$ and $y^{rcnn}$ with (9) (10); 
 return $y$ ; |

**Laplace approximation** The log-posterior weight distribution $\log p(\theta|\mathbf{X},\mathbf{Y})$ can be unfolded with Bayes formula: $\log p(\theta|\mathbf{X},\mathbf{Y}) = \log p(\mathbf{Y}|\mathbf{X},\theta) + \log p(\theta) - \log p(\mathbf{Y}|\mathbf{X})$. Laplace approximation unfolds it at $\theta^*$ ($\nabla_\theta \log p(\theta|\mathbf{X},\mathbf{Y})|_{\theta^*} = \mathbf{0}$) with a second-order Taylor expansion to construct a normal distribution approximating $p(\theta|\mathbf{X},\mathbf{Y})$, i.e.

$$\log p(\theta|\mathbf{X},\mathbf{Y}) \approx \log p(\theta^*|\mathbf{X},\mathbf{Y}) + \frac{1}{2}(\theta - \theta^*)^T \mathbf{H}_{\log p(\theta^*|\mathbf{X},\mathbf{Y})}(\theta - \theta^*), \tag{2}$$

where $\mathbf{H}_{\log p(\theta^*|\mathbf{X},\mathbf{Y})} = \frac{\partial^2}{\partial^2\theta}\log p(\theta^*|\mathbf{X},\mathbf{Y})$ is the Hessian matrix of $\log p(\theta^*|\mathbf{X},\mathbf{Y})$. It is easy to get $\mathbf{H}_{\log p(\theta^*|\mathbf{X},\mathbf{Y})} = \mathbf{H}_{\log p(\mathbf{Y}|\mathbf{X},\theta)} + \frac{\partial^2}{\partial^2\theta}[\log p(\theta)]$, where the second term can be evaluated easily if $p(\theta)$ is simple or in a closed form. The Hessian matrix in the first term can be approximated with the Fisher information matrix [15, 17, 38], which is defined as

$$\mathbf{F} \doteq \mathbb{E}_{P_\mathbf{X}}[\mathbb{E}_{P_{\mathbf{Y}|\mathbf{X},\theta}}[\nabla \log p(\mathbf{Y}|\mathbf{X},\theta)^T \nabla \log p(\mathbf{Y}|\mathbf{X},\theta)]] \tag{3}$$

The approximate posterior weight distribution is $p(\theta|\mathbf{X},\mathbf{Y}) \approx \mathcal{N}(\theta;\theta^*,\Sigma)$, where $\Sigma = [\mathbf{F}+\Sigma_0^{-1}]^{-1}$, if we consider a Gaussian weight prior $p(\theta) = \mathcal{N}(\theta;\mathbf{0},\Sigma_0)$.

## 3.2 Bayesian inference in detection

3D object detectors contain both classification and regression predictions. We conduct Bayesian inference and estimate $p(\mathbf{y}^*|\mathbf{x}^*,\mathbf{X},\mathbf{Y})$ in (1) with Monte-Carlo estimators as [39]. In classification, $p(\mathbf{y}^{*,c} = 1|\mathbf{x}^*,\mathbf{X},\mathbf{Y}) \approx \frac{1}{T}\sum_{\theta \in P(\theta|\mathbf{X},\mathbf{Y})} \text{softmax}^c(f_\theta(\mathbf{x}^*))$, where $p(\mathbf{y}^*|\mathbf{x}^*,\mathbf{X},\mathbf{Y})$ is assumed a categorical distribution, and $T$ denotes the number of Monte-Carlo samples. In regression, we assume $p(\mathbf{y}^*|\mathbf{x}^*,\mathbf{X},\mathbf{Y})$ as a normal distribution. Its mean and covariance matrix can be estimated with

$$\mathbb{E}_{y^*}[\mathbf{y}^*] = \frac{1}{T}\sum_{\theta \in P(\theta|\mathbf{X},\mathbf{Y})} f_\theta(\mathbf{x}^*) \tag{4}$$

$$\text{Cov}[\mathbf{y}^*] = \frac{1}{T}\sum_{\theta \in P(\theta|\mathbf{X},\mathbf{Y})} f_\theta^T(\mathbf{x}^*)f_\theta(\mathbf{x}) + \tau^{-1}\mathbf{I} - \mathbb{E}_{y^*}[\mathbf{y}^*]^T \mathbb{E}_{y^*}[\mathbf{y}^*]. \tag{5}$$

For conciseness, we denote this moment estimation step with MMT-EST$(\cdot)$.

One-stage detectors are end-to-end neural networks, as shown in Figure 2 (a). It outputs two sibling predictions $\mathbf{y}^{cls} \in \mathbb{R}^{M \times C}$ and $\mathbf{y}^{reg} \in \mathbb{R}^{M \times D}$, where $M$ denotes the number of all 3D bounding-box proposals (anchors), $C$ is the number of classes, and $D$ is the dimension of a parameterized 3D bounding box. Algorithm 1 shows the MCBI with moment estimation to estimate predictive distributions $y$. It is note that the set of 3D bounding-box predictive distributions $y$ contains two components: a set of 3D bounding-box predictive classification distributions $y^{cls}$ and its regression counterpart $y^{reg}$, i.e. $y = \{y^{cls}, y^{reg}\}$.

**Mis-matching in two-stage detectors** Two-stage detectors are not end-to-end neural networks and contain two separate sub-networks: RPN and RCNN. RPN generates proposals $\mathbf{y}^{rpn} = f_{rpn}(\theta_{rpn},\mathbf{x})$. A non-maximization-suppression (NMS) module computes the overlapped area among proposals and selects the top-K unoverlapped proposals by their classification scores. We

denote this step as $S_p = \text{NMS-TopK}(\mathbf{y}^{rpn})$, where the $S_p$ contains the indices of selected proposals. Since the ranking results of NMS depending on the RPN output $f_{rpn}(\theta_{rpn}, \mathbf{x})$, different weight samples $\theta_{rpn}$ drawn in Monte-Carlo sampling may generate different selections $S_p$. Formally,

$$S_{p_1} = \text{NMS-TopK}(f_{rpn}(\theta_{rpn}^1, \mathbf{x})), \theta_{rpn}^1 \sim p(\theta_{rpn}|\mathbf{X}, \mathbf{Y}), \tag{6}$$

$$S_{p_2} = \text{NMS-TopK}(f_{rpn}(\theta_{rpn}^2, \mathbf{x})), \theta_{rpn}^2 \sim p(\theta_{rpn}|\mathbf{X}, \mathbf{Y}), \tag{7}$$

$$P(S_{p_1} = S_{p_2}) < 1 \text{ , if } \theta_{rpn}^1 \neq \theta_{rpn}^2. \tag{8}$$

These selected proposals across different weight samples are probably not matched. As a result, the moment estimation process in RCNN cannot estimate the true moments, as shown in Figure 1 (a) (c).

**Uncertainty Separation and Aggregation (U-SPA)**    To solve this problem, we separate the uncertainty estimation for RPN and R-CNN in two-stage detectors, and aggregate them in the final predictive distribution as shown in Figure 1 (b). We conduct MCBI on RPN to obtain the predictive distributions of RPN results $y^{rpn}$. We then extract the mode $\mathbf{y}^{rpn}$ and conduct NMS to get selected proposals $S_p$. In estimating uncertainties in RCNN, we adopt the selected modes $\{\mathbf{y}_i^{rpn}|i \in S_p\}^3$ as inputs and conduct MCBI. With the same input, the outputs of Monte-Carlo samples will be matched in the order defined by $S_p$.

In aggregating the uncertainties from RPN and R-CNN, we keep the mode of $y^{rcnn}$ and accumulate the covariance matrices of $y^{rpn,reg}$ and $y^{rcnn,reg}$. It can be considered as adding up uncertainties from two independent sources. Formally, the final predictive distribution with aggregated uncertainties can be formulated as

$$[y^{cls}]_i = \text{categorical}(\overline{[y^{rcnn,cls}]_i}), i \in S_p, \tag{9}$$

$$[y^{reg}]_i = \mathcal{N}(\overline{[y^{rcnn,reg}]_i}, \Sigma_{[y^{rpn,reg}]_i} + \Sigma_{[y^{rcnn,reg}]_i}), i \in S_p, \tag{10}$$

where the $[\cdot]_i$ denotes the $i$-th element in a set, $\overline{(a)}$ extracts the mode of a predictive distribution $a$, and $\Sigma_a$ extracts its covariance matrix. Algorithm 2 summarizes the U-SPA pipeline.

### 3.3   Diagonal Fisher approximation

In 3D detectors, the number of parameters could be in the order of a million. It is hard to compute the exact Fisher and its inversion directly. To make the Fisher calculation feasible, diagonal Fisher approximation assumes each single weight parameter as independent:

$$\text{diag}(\mathbf{F}) = \mathbb{E}_{P_\mathbf{X}}[\mathbb{E}_{P_{\mathbf{Y}|\mathbf{X},\theta}}[\text{diag}(\nabla \log p(\mathbf{y}|\mathbf{x}, \theta))^2]], \tag{11}$$

where $\text{diag}(\cdot)$ converts a matrix or a vector to its corresponding diagonal matrix. To evaluate standard diagonal Fisher (11), we adopt the Monte-Carlo method used in [38] to calculate the inner expectation over $P_{\mathbf{Y}|\mathbf{X},\theta}$:

$$\text{diag}(\mathbf{F}) \approx \mathbb{E}_{P_\mathbf{X}}[\frac{1}{T_f} \sum_{\mathbf{y} \in P_{\mathbf{Y}|\mathbf{X},\theta}} [\text{diag}(\nabla_\theta \log p(\mathbf{y}|\mathbf{x}, \theta))^2]], \tag{12}$$

where $T_f$ denotes the number of Monte-Carlo samples in calculating the Fisher. It requires $O(NT_f)$ backward computations to calculate the diagonal standard Fisher and $O(P)$ storage space.

**Empirical diagonal Fisher**    The empirical Fisher was adopted as a low-cost surrogate of the standard Fisher or Hessian in [19–21]. It is defined as

$$\tilde{\mathbf{F}} \doteq \frac{1}{N} \sum_{(\mathbf{x},\mathbf{y}) \in \mathcal{D}} [\nabla \log p(\mathbf{y}|\mathbf{x}, \theta)^T \nabla \log p(\mathbf{y}|\mathbf{x}, \theta)]. \tag{13}$$

If we approximate the empirical Fisher with diagonal approximation, we have the following empirical diagonal Fisher formulation:

$$\text{diag}(\tilde{\mathbf{F}}) = \frac{1}{N} \sum_{(\mathbf{x},\mathbf{y}) \in \mathcal{D}} [\text{diag}(\nabla \log p(\mathbf{y}|\mathbf{x}, \theta))^2]. \tag{14}$$

Compared to the standard Fisher information matrix, it avoids the expectation over the predictive distribution and computes the expectation over the training distribution instead of the predictive distribution. Therefore, it only requires $O(N)$ backward computations. Even though the empir-

---

³We denote $\mathbf{y}_i^{rpn} = \{[\mathbf{y}^{rpn,cls}]_i \in \mathbb{R}^C, [\mathbf{y}^{rpn,reg}]_i \in \mathbb{R}^D\}$, the $[\cdot]_i$ returns the i-th element from its input.

| | PP | | | | SC | | | | PR | | | |
|---|---|---|---|---|---|---|---|---|---|---|---|---|
| | classification | | regression | | classification | | regression | | classification | | regression | |
| methods | mean | std.$\times 10^{-4}$ | mean | std.$\times 10^{-3}$ | mean | std.$\times 10^{-4}$ | mean | std.$\times 10^{-3}$ | mean | std.$\times 10^{-4}$ | mean | std.$\times 10^{-3}$ |
| det. | 0.115 | - | - | - | 0.100 | - | - | - | 0.113 | - | - | - |
| dropout | 0.112 | 1.4 | 0.763 | 0.7 | 0.126 | 2.1 | 0.594 | 1.5 | 0.104 | 10.7 | 0.746 | 2.6 |
| deep-en. | 0.106 | 0.0 | 0.664 | 0.0 | 0.096 | 0.0 | 0.640 | 0.0 | 0.608 | 0.0 | 0.767 | 0.0 |
| LL (E.) | 0.115 | 0.5 | 0.528 | 3.6 | 0.100 | 1.0 | 0.510 | 2.2 | 0.111 | 18.9 | 0.357 | 3.2 |
| LM (E.) | - | - | - | - | - | - | - | - | 0.064 | 16.9 | **0.316** | 2.5 |
| full (E.) | **0.097** | 10.2 | 0.418 | 2.3 | **0.087** | 5.2 | 0.395 | 5.1 | **0.061** | 11.9 | 0.576 | 9.6 |
| LL (S.) | 0.115 | 0.8 | **0.302** | 6.2 | 0.100 | 0.4 | **0.300** | 22.8 | 0.111 | 19.1 | 0.519 | 0.4 |
| LM (S.) | - | - | - | - | - | - | - | - | 0.095 | 15.0 | 0.467 | 1.7 |
| full (S.) | 0.111 | 1.6 | 0.338 | 7.7 | 0.089 | 6.5 | 0.311 | 14.3 | 0.070 | 11.8 | 0.481 | 34.8 |

Table 1: Expected calibration error ($\downarrow$). The "det", "LL", "LM", and "full" are short for "deterministic", "last-layer", "last-module", and "full-net". The results are computed from 5 random seeds.

| | PP | | SC | | PR | |
|---|---|---|---|---|---|---|
| methods | mean | std. | mean | std. | mean | std. |
| dropout | 3.055 | 0.604 | 0.185 | 0.011 | 1.435 | 0.175 |
| deep-en. | 0.104 | 0.000 | 0.043 | 0.000 | 1.546 | 0.000 |
| LL (E.) | 0.332 | 0.021 | 0.193 | 0.019 | -0.599 | 0.007 |
| LM (E.) | - | - | - | - | **-0.690** | 0.007 |
| full (E.) | **-0.744** | 0.002 | **-0.773** | 0.006 | -0.435 | 0.011 |
| LL (S.) | -0.561 | 0.022 | -0.603 | 0.079 | -0.007 | 0.012 |
| LM (S.) | - | - | - | - | -0.297 | 0.011 |
| full (S.) | -0.666 | 0.005 | -0.746 | 0.021 | -0.574 | 0.041 |

Table 2: Negative Log-likelihood ($\downarrow$). The notations are the same to Table 1.

ical Fisher has superior computation efficiency and practical success, it lacks a solid theoretical grounding to capture the local curvature information[18]. In Section 4, we will compare the Laplace approximation with standard diagonal Fisher and empirical diagonal Fisher in 3D object detection.

# 4   Experiments

In this section, we conduct experiments on the KITTI dataset [1] to explore the following questions: (1) How is the quality of the estimated predictive distribution? (2) Does the accuracy of mode estimation get affected after applying Laplace approximation? We adopt well-trained 3D object detectors as deterministic models and convert them with the Laplace-approximation-based approaches in Section 3. The adopted 3D detectors contain one-stage detectors PointPillar[25] ("PP") and Second[23] ("SC"), as well as a two-stage detector PointRCNN[27] ("PR"). In particular, we compute the Fisher on the training set and inference with it on the validation set. Ten Monte-carlo samples are adopted to approximate the integration in (1). We induce $\alpha$ as the scalar factor of the posterior weight covariance matrix, and set $\alpha = 0.003$ for the standard Fisher, and $\alpha = 0.01$ for the empirical Fisher by tuning it on the hand-hold set.

## 4.1   Quality of predictive distributions

In this section, we present the Expected Calibration Error (ECE) and Negative Log-Likelihood (NLL) to demonstrate the predictive distribution quality, compare the following baselines and various Laplace-approximation implementations:

- deterministic model: we use the softmax outputs as classification confidences;
- mc-dropout [39]: we add a dropout layer before the last convolution/linear layer to generate the Monte-Carlo-based predictive distribution. To make a fair comparison, we use the same model parameters as the others without retraining;
- deep-ensembles [40]: we train the same network independently with different initialization $T$ times, and ensemble them to generate the predictive distribution. The deep-ensembles (PR) adopts the proposed U-SPA pipeline to make it work in practice;
- full-net (E./S.): we implement Laplace approximation on all the layers with the diagonal approximation of the diagonal empirical/standard Fisher;
- last-layer (E./S.): Laplace approximation is applied on the last-layer only;
- last-module (E./S.): Laplace approximation is applied on the R-CNN network.

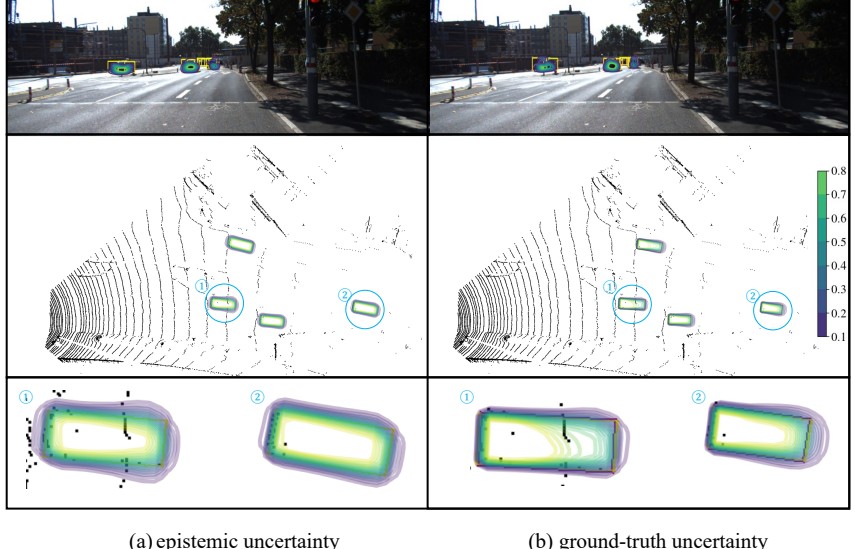

(a) epistemic uncertainty
(Laplace approximation)

(b) ground-truth uncertainty
(generative model [34])

Figure 3: Visualization of estimated epistemic uncertainty (left), and the ground-truth spatial uncertainty [34] used in evaluating mAP(JIoU) (right). The top, middle, and bottom rows are front, bird's-eye, and zoom-in views. The contours reflect the objectiveness probability.

Table 1 and Table 2 report the ECE and NLL results. The estimated predictive distributions are better calibrated than deterministic model outputs. Full-net Laplace approximation perform consistently better than MCDropout and DeepEnsembles baselines, if we compare the rows of full(E./S.) with dropout and deep-en. in Table 1. MCDropout models the posterior weight distribution of a subset of parameters with a Bernoulli distribution. It probably cannot characterize the real posterior. DeepEnsembles adopt $T$ independently-trained optimal weight points to generate the predictive distribution. It can model the multi-modality in weight posterior, but each weight point is a deterministic model and suffers from poor calibration.

We have two interesting findings. Firstly, the empirical Fisher results perform competitive results to the standard Fisher counterparts in Table 1 and 2, though it lacks theoretical foundations. We attribute this to the fact that we calculate the empirical Fisher at the optimal $\theta^*$, which makes the predictions similar to the ground-truth labels. Secondly, an increasing classification calibration quality can be seen when Laplace approximation is applied to increasing number of layers, if we compare LL, LM, full in Table 1. The same trend can also be seen in the NLL results (Table 2). It is opposite to the finding in [11], which claims the last-layer empirical Fisher owns the best performance and full-net Fisher suffers from an under-fitting problem. We attribute our result to the fact that we implement the Laplace approximation with a diagonal approximation which contains much fewer parameters than the either full or K-FAC Fisher matrix used in [11].

## 4.2   Accuracy of estimated mode

In this section, we explore the effect of Laplace approximation on detection accuracy by varying $\alpha$, which is the scalar factor of the posterior weight covariance matrix. When $\alpha = 0$, the posterior weight distribution will degenerate to a deterministic model. When $\alpha$ increases, the posterior weight distribution increasingly spans the weight space. We calculate the average mAP(IoU) over all classes and different levels denoted as $\mathbb{E}_{\text{class,level}}[\text{mAP(IoU)}]$ in Figure 4. It is found that as $\alpha$ increases, $\mathbb{E}_{class,level}[\text{mAP(IoU)}]$ decreases. It achieves its maximum at $\alpha = 0$, in which the model is deterministic and cannot estimate uncertainty. The mAP(IoU) degrades slightly when the $\alpha$ is small, while an extremely large value of $\alpha$ will greatly deteriorate the mAP(IoU). It suggests adopting a relatively small value of $\alpha$ to obtain the uncertainty and preserve the mAP(IoU) performance.

## 4.3   Visualization and discussion on predictive distributions

To better understand the estimated epistemic uncertainty, we visualize our estimated predictive distributions compared to those estimated with a generative model in [34]. The generative model characterizes the generative process of key-object points within 3D bounding boxes. It generates the predictive distribution with ground-truth 3D bounding box annotations and LiDAR observations.

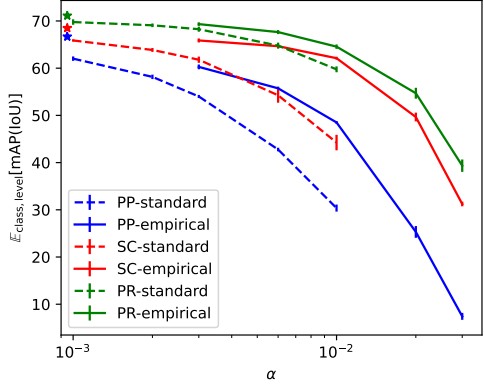

Figure 4: Relationship between $\alpha$ and $\mathbb{E}_{\text{class,level}}[\text{mAP(IoU)}]$ ($\uparrow$) (full-net). The stars depict the results of the deterministic models. The results are computed from 5 random seeds.

| method | class | bev | | | 3d | | |
| --- | --- | --- | --- | --- | --- | --- | --- |
| | | easy | mod. | hard | easy | mod. | hard |
| PP | car | 80.2 | 78.9 | 75.5 | 43.7 | 28.0 | 23.5 |
| | ped. | 19.2 | 20.8 | 20.7 | 29.9 | 29.8 | 27.6 |
| | cyc. | 48.3 | 42.6 | 40.3 | 46.3 | 33.0 | 31.2 |
| SC | car | 87.0 | 84.8 | 81.3 | 58.2 | 37.6 | 31.3 |
| | ped. | 23.6 | 26.2 | 25.7 | 30.9 | 30.9 | 28.2 |
| | cyc. | 50.6 | 48.3 | 46.4 | 45.5 | 34.7 | 33.1 |
| PR | car | 81.1 | 78.8 | 75.1 | 41.4 | 29.9 | 26.1 |
| | ped. | 26.0 | 25.9 | 24.0 | 39.4 | 35.8 | 31.3 |
| | cyc. | 62.7 | 51.9 | 49.4 | 61.3 | 43.1 | 40.7 |

Table 3: mAP(JIoU) (standard Fisher, full-net) results on the KITTI *val* set .The reported results are mean values from 5 random seeds.

We project the predictive distributions to the bird's-eye-view (BEV) image plane and get objectiveness probability for visualization. The objectiveness probability is the probability of a BEV region subject to a key object, like cars, pedestrians, and cyclists. Figure 3(b) shows the objectiveness probability of the ground-truth generative model. The objectiveness probability of the ground-truth generative model is high around key objects. It is relatively low in the region with sparse LiDAR observations. We visualize estimated predictive distributions of Laplace approximation in Figure 3(a). It shows similar behavior to the ground-truth generative model in Figure 3(b), and confident in dense-point regions and relatively uncertain when the LiDAR observation is sparse.

Table 3 reports the mAP(JIoU) between predictive distributions estimated with Laplace approximation and those with the generative model. It shows that the estimated epistemic uncertainty behaves like the generative model in relatively large objects, like cars and cyclists.

## 5 Limitations

The proposed method computes the Bayesian inference formula based on Monte-Carlo, It requires $T\times$ computational costs more than a single forward computation. The run-time performance can be improved if it parallels the Monte-Carlo computations with batched inputs [39]. For the robotic application with strong constraints on computational resources, the subnet implementation that only applies Laplace approximation on the last layer also provides good-quality uncertainty estimations, as shown in Table 1. This paper assumes that both weight and predictive distributions conform to the normal distribution. It is limited in characterizing the multi-modality. It can be partially alleviated by fitting Laplace approximation to each modality [41].

## 6 Conclusion

In this paper, we tailor Laplace approximation for 3D object detectors. An uncertainty-separation-and-aggregation method is proposed to make it work in two-stage 3D object detectors. The experiment results on the KITTI dataset show that the estimated epistemic uncertainty is of better quality than deterministic models and Monte-Carlo Dropout baselines. Even though it lacks theoretical foundations, empirical diagonal fisher shows competitive performance to standard diagonal fisher in Laplace approximation. Full-net diagonal Laplace approximation, which provides good uncertainty quality, is recommended in offline applications, like model diagnosis, active learning, and life-long learning. If the robotic applications have strong computational resource constraints, subnet Laplace approximation is a good balance between run-time performance and uncertainty quality. In the future, it will be interesting to overcome the discussed limitations and explore the usefulness of the estimated epistemic uncertainties in downstream tasks, like path planning.

**Acknowledgments**

This work was supported by Guangdong Basic and Applied Basic Research Foundation, under project 2021B1515120032, Zhongshan Science and Technology Bureau Fundunder project 2020AG002, and Project of Hetao Shenzhen-Hong Kong Science and Technology Innovation Cooperation Zone(HZQB-KCZYB-2020083), awarded to Prof. Ming Liu.

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
