# OpenReview forum: "Laplace Approximation Based Epistemic Uncertainty Estimation in 3D Object Detection"
_robot-learning.org/CoRL/2022/Conference — CoRL 2022 Poster_

### Official Review · Reviewer_mzVN · 2022-07-04

**Originality:** Fair
**Technical Quality:** Good
**Clarity Of Presentation:** Good
**Impact:** 3

**Recommendation:**

Weak Accept: I recommend accepting the paper, but will not argue for my recommendation if the majority of other reviewers have a different opinion.

**Summary:**

The authors present a method of applying Laplace approximation (LA) to one- and two-stage 3D object detectors for epistemic uncertainty estimation. In the two-stage case, LA cannot directly be applied as some steps in the pipeline are non-differentiable. The presented solution consists of individual application of LA to both stages and an aggregation step. Further, a heuristic for prior estimation, i.e. regularization of LA, is established through experimentation. In a comparison to a deterministic and a Monte Carlo Dropout baseline on the KITTI dataset, superiority in uncertainty quality, i.e. calibration, is established.

**Issues:**

* Clear distinction between Hessian, Fisher and GNN.
* Potential comparison to more expressive Posteriors than isotropic Gaussian like KFAC.
* Influence of number of MC samples during inference.
* Further evaluation of uncertainty quality going beyond calibration. Inclusion of additional metrics like NLL or KL-divergence.
* Clearer discussion of benefits of uncertainty estimation for the 3D object detection case (autonomous driving and robotics). What is possible when using the proposed approach that was not possible before?
* Inclusion of future work and code.

**Quality Of The Limitations Section:**

Limitations are addressed clearly

**Reviewer Expertise:**

4: The reviewer is confident but not absolutely certain that the evaluation is correct

**Robotics Focus:**

Relevant but unlikely to deploy to hardware in near future

**Strengths And Weaknesses:**

## Strength
* Practical approach to uncertainty estimation in 3D deep neural networks (DNNs).
* Detailed and largely consistent exposition (e.g. LA background section).
* Helpful algorithm summaries.
* Comparison of empirical and standard Fisher performance.
* Comparison of varying degrees of inclusion of layers into the LA computation.
* Insightful discussion of limitations.

## Weaknesses
* Lack of originality. The application of LA to 3D object detection is rather straight forward and mostly combines existing algorithms and model architectures.
* The Fisher information matrix is not merely a "good substitute" (line 119) to the Hessian but equivalent to it given piecewise linear activation functions and exponential family loss functions (this does not hold for the empirical Fisher).
* Missing distinction between the Fisher and Generalized Gauss-Newton (GNN) matrix which are only equivalent in the special case of exponential family loss functions.
* Ad-hoc method for posterior regularization.
* ECE as sole metric for uncertainty quality quantification and missing discussion of its limitations. NLL might be a worthwhile inclusion in the regression case.
* Uncertainty quality is reduced to calibration. Separation of out-of-distribution examples (KL-divergence) could be included.
* Only diagonal LA.
* Missing code.
* No discussion of future work.
* Slight language issues, e.g.:
  * _admirable_ vs _desirable_ (line 1)
  * _indifferentiable_ vs. _non-differentiable_ (line 4)
  * _adorable_ vs _useful/desirable_ (line 62)

**Summary Of Recommendation:**

While the paper presents a practical approach to uncertainty estimation in 3D object detection which is quite relevant for robotics applications, missing originality and lacking experimental evaluation and discussion hold it back.

---

> ### Author Response · Authors · 2022-08-22
> **Response to Reviewer-mzVN (Updated)**
>
> # Reviewer mzVN
>
> The authors would like to thank these valuable comments from Reviewer-mzVN. Following contains the response to the comments. Two pdf files are included to clarify these points:
>
> - appendix file (**updated in Aug26**)
> - revised paper (**updated in Aug26**)
>
> The updated responses are marked with **updated in Aug26**.
>
> ## Regarding "Lack of originality."
>
> This paper applies the existing Laplace approximation and its variances to 3D object detection, which is not straightforward due to its high dimensional output space and non-end-to-end architecture.
> It provides solutions to achieve a tractable Laplace-approximation-based 3D probabilistic object detection pipeline for both RPN-based and RCNN-based 3D detectors, which cover the most popular detectors in autonomous driving applications.
> Both evaluation protocol, baselines, and code (attached in the supplementary) can boost the community to explore probabilistic object detection. They will encourage downstream tasks using probabilistic perceptual results instead of deterministic ones.
>
> ## Regarding "The Fisher information matrix...the empirical Fisher)."
>
> **updated in Aug26**
>
> The "substitute" has been fixed in the revised version.
> Indeed, empirical Fisher is not equivalent to the Hessian, and we have discussed it in Line172-175 of the revised paper.
> Hope it can clarify this point.
>
> ## Regarding "Missing distinction between...family loss functions."
>
> Thanks for the comment. The descriptions on "GGN" might make readers confused.
> Both Fisher and GGN can be an approximation of the Hessian matrix. [Martens2020JMLR] provides a way to calculate the Fisher exactly (Section 9.1 equation (7) in [Martens2020JMLR], and equation (10) in our original paper), which is similar to the GGN approximation equation.
> It is distinct from GGN approximation that the inner expectation of Fisher calculation $E_{P_{Y|X, \theta}} [ \nabla_{f_\theta(x)} \log p(y|f_\theta(x))^T \nabla_{f_\theta(x)} \log p(y|f_\theta(x))]$ is the fisher of the predictive distribution instead of Hessian matrix $H_L = \frac{\partial^2}{\partial^2 f_\theta(x)} \log p(y|f_\theta(x))$ of the log posterior w.r.t. the network output $f_\theta(x)$.
>
> In our original paper, the "GGN-based Fisher calculation approach" denotes the exact computation way of calculating the Fisher instead of the GGN approximation. To avoid confusion, we have changed the phrase "GGN-based Fisher calculation" into "exact Fisher calculation" in the revised appendix.
>
> [Martens2020JMLR] Martens, James. "New insights and perspectives on the natural gradient method." The Journal of Machine Learning Research 21.1 (2020): 5776-5851.
>
> ## Regarding "Ad-hoc method for posterior regularization."
>
> **updated in Aug26**
>
> The authors are unclear about the "Ad-hoc method for posterior regularization."
>
> Does the reviewer mention the post-hoc posterior approximation?
> In fact, the Laplace approximation method applied to estimate the posterior is a post-hoc way without changing the training procedure. It is an admirable feature that all existing well-trained 3D detectors can be directly used with the Laplace approximation approaches to extend as a probabilistic detector.
>
> Suppose the reviewer thought this weakness was from Section3.1.3. In that case, we have to admit that the solution given in Section3.1.3 is proposed specifically to solve the problem that some elements in the Fisher approximation suffer from intrinsic inaccuracy.
> The section of " A heuristic approach to sampling from weight distributions" is removed from the main paper in the revised version due to this problem and page limitation.
>
> ## Regarding "ECE as sole...the regression case."
>
> **updated in Aug26**
>
> Thanks for the comment. We have reported the NLL performance in Table 2 of the revised paper.
>
> ## Regarding "Uncertainty quality...be included."
>
> It is true that some experiments, like OOD detection, can enhance the experiment part.
> We will add the OOD detection experiment if the time is permitted in this discussion period.
> Sorry for not providing the OOD detection experiment results due to the limited time in this discussion period.
> But it is really good advice to improve our work, and we will leave it in our future work.

---

> ### Author Response · Authors · 2022-08-22
> **Response to Reviwer-mzVN (updated)**
>
>
> ## Regarding "Only diagonal LA"
>
> **updated in Aug26**
>
> The Laplace approximation based epistemic uncertainty pipeline can support various Laplace approximation approaches, not limited to the diagonal LA.
> It would be interesting to compare with K-FAC Laplace approximation, but sorry for not being able to provide K-FAC results in this discussion period...
> We have tried to actuate K-FAC Fisher in our task, but it requires more time to refactor the code and conduct experiments, since 3D detectors contain special operations, like sparse 3D convolution, and existing Fisher calculation packages do not support them well.
>
> To improve the experiment part, we have added the deep ensembles as a baseline in the experiment of the revised paper.
> We train the same network independently with different initialization T
> times, and ensemble them to generate the predictive distribution.
> Compared to DeepEnsembles, Laplace approximation show better or competitive results in our experiments.
>
> ## Regarding "Missing code"
>
> The code was provided in the supplementary material.
>
> ## Regarding "No discussion of future work"
>
> This work provides an applicable pipeline and demonstrates its usefulness with Diagonal Fisher approximation. The future work is to overcome the three limitations discussed in Section5 Limitations of the original paper. In detail:
>
> - Run-time performance: Monte-Carlo-based inference requires $T\times$ computational costs than a single forward computation, where $T$ denotes the number of Monte-Carlo samples during inference. It can be improved by parallelling the $T$ Monte-Carlo computations with batched inputs [Gal2016]. A real-time detection system with additional epistemic uncertainty output will be more desirable for downstream tasks.
>
> - Uni-modality assumption: It will be interesting to extend the current uni-modality Laplace approximation to multi-modality counterparts, which is more capable to characterize the posterior distribution of deep neural networks.
>
> - Applications: In this work, we demonstrate the superior calibration performance of estimated epistemic uncertainty. Its application in downstream tasks, like path-planning and manipulation, or in model enhancement will also be interesting directions to explore in the future.
>
> We have added future work to the conclusion of the revised paper.
>
> ## Regarding "Slight language issues"
>
> Thanks for the comments. They have been fixed in the revised paper.
>
> ## Regarding "Clear distinction between Hessian, Fisher and GGN"
>
> It has been replied in [Regarding "Missing distinction between ... exponential family loss functions.](#Regarding-to-"The-Fisher-information-...-empirical-Fisher").
> This problem has been fixed in the revised paper and appendix.
>
> ## Regarding "Clearer discussion of benefits of uncertainty estimation for the 3D object detection case (autonomous driving and robotics). What is possible when using the proposed approach that was not possible before?"
>
> **Autonomous driving**: The estimated epistemic uncertainty for 3D object detectors can benefit downstream tasks in autonomous driving, like path planning. Currently, the path planning module generates paths in terms of deterministic perceptual results without considering their uncertainties. With the additional epistemic uncertainty, the path-planning module can output a more conservative path for actuators when the perceptual module detects objects with high uncertainty in its surroundings.
>
> ## Regarding "Inclusion of future work and code"
>
> **Future work**: It has been replied in [Regarding-"No discussion of future work"](#Regarding-to-"No discussion-of-future-work")
>
> **Code**: The code was provided in the supplementary material.
>
> ## Regarding "Influence of number of MC samples during inference."
>
> **updated in Aug26**
>
> We report the influence of $T$ on the expected error in Section8 of the attached appendix.
>
> ## "Further evaluation of uncertainty quality going beyond calibration. Inclusion of additional metrics like NLL or KL-divergence."
>
> Thanks for the comment. We have reported the NLL performance in Table 2 of the revised paper.

---

### Official Review · Reviewer_z9CZ · 2022-07-20

**Originality:** Fair
**Technical Quality:** Good
**Clarity Of Presentation:** Good
**Impact:** 3

**Recommendation:**

Weak Reject: I recommend rejecting the paper, but will not argue for my recommendation if the majority of other reviewers have a different opinion.

**Summary:**

This work proposed to estimate the epistemic uncertainty of a deep 3D object detector by treating the deep neural net as a Bayesian Neural Network (BNN) whose weight posterior is inferred by Laplace approximation(LA).  In order to conquer different issues such as high-dimensional output space and in-differentiable operations during the adaptation, the authors employed the "standard Fisher" approximation along with the empirical one and introduced a moment matching processes to collect uncertainty from different modules. Besides, a heuristic method to select the hyper-parameters of the Covariance matrix has been adopted to boost the performance. In experimental validation, the proposed approach is demonstrated better performance over deterministic networks and Monte-Carlo dropout on KITTI dataset, with the metric of expected calibration error for both classification and regression. The performance of mAP (JIoU), another metric for predictive distribution, is also reported.

**Issues:**

As mentioned in the section of strengths and weakness, if the majority of the aforementioned points can be addressed, the reviewer would consider to raise the score of this work.

**Quality Of The Limitations Section:**

Additional details required

**Reviewer Expertise:**

3: The reviewer is fairly confident that the evaluation is correct

**Robotics Focus:**

Relevant but unlikely to deploy to hardware in near future

**Strengths And Weaknesses:**

Overall, the reviewer appreciates the authors' work because it is a quite relevant and urgent topic for robotic perception in safety-critical applications. Though there has been a plethora of advances in uncertainty estimation within the field of machine learning, to adapt them on more practical tasks and validate their effectiveness is also imperative and of greater importance. Besides the topic relevance, the presentation is generally clear and well-organized, easy to follow though with some flaws mentioned later. Nevertheless there is room to improve in terms of presentation, algorithms and experiments from reviewer's perspective. The reviewer hopes that the following suggestions are helpful on improving the quality of the paper.

#### Presentation
- the main text spends much space on introducing the full Laplace Approximation with less focus on the diagonal variant which is the one used finally. This gives the reader a misleading impression. Though there are motivations such as memory issue for only using the diagonal variant but they have not been discussed thoroughly.
-  it would be more clear to move the introduction of LA in section 3 into a separate section for preliminaries or background; Further this section would be better to include a brief description of the commonly used architecture for 3D object detectors. Currently, there is no definition of RPN and R-CNN except for their appearance in the related work section.
- regarding two different ways of computing the Fisher (empirical and standard) in section 3.1.1, it would be more clear to put the equations of both ways in the main text since this is one of the relevant parts of the method.
- it would be much more concise if the authors can use equations to describe the uncertainty separation and aggregation module instead of Fig.2 which has many undefined letters and such as N, W, L, M and so on. And this figure is not a bit too complex in communicating the main idea.
- better to explain with equation for section 3.1.3;
- experiments:
     - in Table 1 `std.` is ambiguous since it can also mean standard deviation.
     - in Fig.3 the figure size is too small, while the width of the curves is large.
     - in Fig.5 there is no `stars` in the figure but it is mentioned in the caption.

#### Algorithms
-  the foremost concern is that, the authors highlight the problem of high-dimensional output space when applying LA and suggest a Monte-Carlo approximation for standard Fisher (evaluation over the expectation of predictive distribution) to get around the intractable analytical form, which is fine. But this is not the problem of empirical Fisher (evaluation over the expectation of data distribution). And in the experiments, the results show that empirical can have better performance than the standard one. The question here is, why don't the authors just use the empirical version? What's the main motivation of employing standard Fisher? This is a significant but very interesting point. It would be great if the author can explain the main difference in details between these two, and try to validate the insights from the previous works in theory and verify if these insights are reflected in a more practical setting or if they are helpful in guiding the design for a more effective algorithm.

- The second concern is still around LA. It's understandable to use diagonal variant only with the motivation of resources-restricted requirements in robotics. But the problem is that,  what kind of role does the relation between weights play in tasks like 3D object detection, indispensable or ignorant? Since the run-time problem from sampling is anyway there, therefore there is no difference on this aspect for the variants with richer relation modeling between weights such as K-FAC [1]. The authors mentioned this in the related work but neglect it without a persuasive argumentation.

- Regarding the uncertainty separation and aggregation, there are still in-differential operations such as ranking and selection in the proposed module, how does the information flow back and differ with the old model? Why to use addition for uncertainty aggregation instead of other operations such as multiplication or maximum and so on?

- What's the main difference between the heuristic approach in section 3.1.3 and hyper-parameter tuning based on validation set?

#### Experiments
- As mentioned before in the algorithm part, the reviewer would like to see the comparison with some more generally accepted baseline such as deep Ensemble [3]; It would also be great to compare with K-FAC [1] or at least to provide more arguments why it's not considered in this task.

- In table 4, it's a little unreasonable to have zero standard deviation in classification for PP and SC. Have the authors tried to vary different number of samples drawn from the weight posterior?

- It would be better to have a ablation study on the uncertainty separation and aggregation module, which can provide more evidence that this module can work and the analysis of the mechanism behind is valuable as well.

- It would be better to provide comparison against other baselines in experiment 4.3.

#### Limitation

- Besides the parallel computation for sampling, some approach on uncertainty propagation such as [2] can be used to mitigate the run-time inefficiency problem.


[1] Martens, James, and Roger Grosse. "Optimizing neural networks with kronecker-factored approximate curvature." International conference on machine learning. PMLR, 2015.

[2] Postels, Janis, et al. "Sampling-free epistemic uncertainty estimation using approximated variance propagation." Proceedings of the IEEE/CVF International Conference on Computer Vision. 2019.

[3] Lakshminarayanan, Balaji, Alexander Pritzel, and Charles Blundell. "Simple and scalable predictive uncertainty estimation using deep ensembles." Advances in neural information processing systems 30 (2017).

**Summary Of Recommendation:**

According to the strengths and weakness section, the reviewer would like to weakly reject this work at current status. Though the overall idea is relevant and interesting, the lack of clear motivation in technical details and evident experimental evaluations on sub-components, data sets and with other baselines constitutes the main concern for the reviewer when assessing this work.

---

> ### Author Response · Authors · 2022-08-22
> **Response to Reviewer z9CZ (updated)**
>
> The authors would like to thank these valuable comments from Reviewer-z9CZ. Following contains the response to the comments. Three pdf files are included to clarify these points:
>
> - revised paper (**updated in Aug26**)
> - appendix file (**updated in Aug26**)
> - response file (**updated in Aug26**)
>
> The updated responses are marked with **updated in Aug26**.
>
> ## Presentation
>
> ### Regarding "the main text...been discussed thoroughly"
>
> Sorry for the misleading impression.
> The diagonal Laplace approximation was written in the appendix.
> In the revised paper, we write the diagonal Fisher and discuss its computation complexity as well as storage consumption in Section3.3.
>
> ### Regarding "it would be...related work section"
>
> **Updated in Aug26**
> Thanks for this comment.
> We add a figure and descriptions to provide the preliminaries of 3D object detection (Section2.2 and Figure2 in the revised paper).
> In addition, the Paragraph of "Mis-matching in two-stage detectors" is added to guide the challenge of applying Laplace approximation in this task in the revised paper.
>
> ### Regarding "regarding two different...of the method."
>
> Thanks for the comments. They have been added in the revised paper (Section3.3).
>
> ### Regarding "it would be...the main idea"
>
> Thanks for the comments. We have revised the Section "Uncertainty Separation and Aggregation" with equations. The Fig.2 in the original paper has been updated with math notations either. Hope it can help communicate the main idea of this part.
>
> ### Regarding "better to explain with equation for section 3.1.3"
>
> **Updated in Aug26**
> In the revised version, we do not include this heuristic sampling method as a major contribution in the main paper, and move the content of this part to the appendix.
> To improve the readability , we have revised this section by separating it into three steps: (1) remove the effects of inaccurate Fisher elements, (2) set a proper prior, and (3) set a proper scaler. Equations are included in Algorithm-A2 to help explain in the revised appendix.
>
> ### Regarding to "experiments:...caption"
>
> Thanks for the comments.
> They have been fixed in the revised paper.
>
> ## Algorithms
>
> ### Regarding "why don't the authors just use the empirical version? What's the main motivation of employing standard Fisher?"
>
> **Updated in Aug26**
> The main motivation for employing standard Fisher is its theoretical foundations which empirical Fisher lacks.
> In prior work, whether employing empirical Fisher is a safe choice in applying Laplace approximation in 3D object detectors is a problem.
> According to our experiment results, the empirical Fisher demonstrates competitive performance as standard Fisher counterparts, which provides empirical evidence for future researchers in employing empirical Fisher in this task.
>
> **Motivation for employing standard Fisher**
> The theoretical foundation of standard Fisher is the major reason we consider it in this paper.
> The empirical Fisher can be the substitution of Hessian and make the calculation tractable.
> But it lacks a theoretical foundation in approximating Hessian, while standard Fisher owns.
>
> **Experiment findings**
> Here we list two experiment findings to support the motivation:
>
> - We cannot get the conclusion that empirical Fisher is more admirable in uncertainty quality,
> since there exist some rows in Table 1 that standard Fisher performs better than empirical Fisher (like SC-LL(stand.) vs. SC-LL(emp.)).
> We can attribute the relatively poor performance of standard Fisher to the Diagonal Fisher approximation, which approximates the Hessian poorly, even with the theoretical foundation.
>
> - If the posterior distribution is estimated precisely, the Bayesian inference over all parameters (full) is expected to provide better uncertainty quality than that over partial parameters (LM and LL). In Table 1 and 2, the uncertainty quality of full-net Laplace approximation is consistently better than sub-net Laplace approximation in standard Fisher results. But empirical Fisher results contain exceptional cases if we compare the rows PR-LL/LM/full(E.). It provides evidence that empirical Fisher approximation suffers from greater intrinsic Fisher inaccuracy than standard Fisher.
>
> ### Regarding "The second concern... a persuasive argumentation"
>
> Since 3D object detectors are deep convolution neural networks or MLP-based networks, diagonal approximation might be poor because of the strong posterior correlations in parameters[MacKay1998].
> The Laplace approximation based epistemic uncertainty in 3D object detection is not limited to diagonal Fisher approximation. The experiment section in the original paper utilizes the diagonal Fisher to demonstrate the feasibility of the proposed Laplace approximation framework in 3D object detection and its superior performance to the MCDropout baseline.
>
> [MacKay1992]MacKay, D. J. (1992). A practical Bayesian framework for backpropagation networks. Neural computation, 4(3), 448-472.

---

> ### Author Response · Authors · 2022-08-22
> **Response to Reviewer-z9CZ (updated)**
>
>
> ### How does the information flow back and differ from the old model?
>
> The ranking in NMS is non-differentiable operations.
> The gradient of the loss computed wth the final output $\mathbf{y}^{rcnn}$ w.r.t. the RPN parameters $\theta_{rpn}$ cannot be computed with chain rule derivatives.
> The proposed "uncertainty separation and aggregation" does not change the original information flow in gradient computation.
> The RPN and RCNN are separately trained in the original PointRCNN.
> Similarly, their Fisher matrices are calculated separately in our method.
>
> The major impact of the non-differentiable operation (ranking in NMS) is that
> it makes different Monte-Carlo weight samples $\theta_{rpn}$ generate different selections $S_p$.
> These selected proposals across different weight samples are probably not matched in numbers or orders. As a result, the moment estimation process in RCNN cannot estimate the true moments.
>
> The "non-differentiable operation" in the original paper might mislead readers. We have clarified this point in Section3.2 of the revised paper.
>
> ### Why to use addition for uncertainty aggregation instead of other operations such as multiplication or maximum and so on?
>
> We adopt the addition operation for uncertainty aggregation due to its theoretical explanation that the addition of covariance matrices can be considered as aggregating two independent Gaussian random variables with the same mean values. It is reasonable since the means of $y^{rpn, reg}$ and $y^{rcnn, reg}$ are similar but with minor refinement. The operation of multiplication and maximum are also applicable in this aggregation step but may lack theoretical explanations.
>
> ### "What's the main difference between the heuristic approach in section 3.1.3 and hyper-parameter tuning based on validation set?"
>
> The major difference is that the heuristic approach determines the weight prior adaptively for each layer. Its comparison (naive method) is a hyper-parameter tuning method in Section3.1.3 of the original paper, which applies a homogenous $\Sigma_0$ for all parameters and tunes a scalar $\alpha$ on the validation set.
>
> ### Regarding "As mentioned before...in this task"
>
> **Updated in Aug26**
>
> Thanks for the comments. We have added the deep ensembles as a baseline in the experiment of the revised paper.
> We train the same network independently with different initialization T
> times, and ensemble them to generate the predictive distribution.
> Compared to DeepEnsembles, Laplace approximation show better or competitive results in our experiments.
>
> It would be interesting to compare with K-FAC Laplace approximation, but sorry for not being able to provide K-FAC results in this discussion period...
> We have tried to actuate K-FAC Fisher in our task, but it requires more time to refactor the code and conduct experiments, since 3D detectors contain special operations, like sparse 3D convolution, and existing Fisher calculation packages do not support them well.
>
> ### Regarding the "zero standard deviation"
>
> **Updated in Aug26**
>
> They do not have zero standard deviation but are extremely small and not represented at that precision level. This problem has been fixed in the revised paper.
> Table 1 of the revised paper represents the standard deviation in scientific notation. It is noted that the std of DeepEnsembles is zero in Table 1 and 2, since it is an ensemble of multiple deterministic models.
>
> ### Have the authors tried to vary different number of samples drawn from the weight posterior?
>
> **Updated in Aug26**
>
> When we tuned the number of MC samples drawn from the weight posterior (denote as $T$), the performance generally improves with increasing value of $T$. To balance speed and performance, we select $T=10$ in our experiments.
> We also report the influence of $T$ on the expected error in Section8 of the attached appendix.
>
> ### Regarding "ablation study on the uncertainty separation and aggregation module".
>
> It is replied in Section1 of the attached response file.
>
> ### Regarding "It would be...in experiment 4.3"
>
> It is reported in Section2 of the attached response file.
>
> ### Regarding "Besides the parallel...run-time inefficiency problem."
>
> Thanks for the comment.
> The error propagation approaches can improve the run-time inefficiency.
> It is very interesting to combine the error propagation idea to improve the run-time inefficiency in the future.

---

> ### Author Response · Authors · 2022-08-22
> **Response file  (updated)**
>
> The attachment contains the response pdf file (**updated in Aug26**).

---

### Official Review · Reviewer_S5g8 · 2022-07-23

**Originality:** Good
**Technical Quality:** Fair
**Clarity Of Presentation:** Poor
**Impact:** 3

**Recommendation:**

Weak Accept: I recommend accepting the paper, but will not argue for my recommendation if the majority of other reviewers have a different opinion.

**Summary:**

This paper presents an epistemic uncertainty estimation technique for 3D object detection. Specifically, a class of approximate Baysian inference method called Laplace Approximation is applied to uncertainty estimation in 3D object detection. The paper suggests three techniques for bringing Laplace Approximation to 3D object detection: (1) a technique of approximating fisher information matrix via monte carlo sampling from pre-specified normal and categorical distributions, (2) a technique of separating uncertainty estimates from region proposal network and RCNN, and (3) a heurstic based approach to weight prior determination. Experiments on KITTI dataset show that the Laplace Approximation performs better than widely used MC-dropout, and different versions of Laplace Approximation is ablated for further empirical insights.

**Issues:**

Other minor comments are:

1. Section 2.3 should change the name to metrics and measures of uncertainty. Moreover, this part is not needed, since there is no contribution of the paper in that direction. I would use that space to discuss existing probabilistic object detection and what advancements the paper is making with respect to that literature.

2. Figure 1: I would use KITTI and realistic object detection architecture to validate this point, rather than using mnist with some linear model. This is because, there are suddle difference between above two set ups, and mnist with linear model does not validate that the fisher matrix converges with enough samples.

3. Line 187-193: what is exact difference between point 1 and point 3? are they both validated with empirical data?

4. Error bars needed for the results, especially table 2 and figure 5.

**Quality Of The Limitations Section:**

Limitations are addressed clearly

**Reviewer Expertise:**

4: The reviewer is confident but not absolutely certain that the evaluation is correct

**Robotics Focus:**

Relevant but unlikely to deploy to hardware in near future

**Strengths And Weaknesses:**

Uncertainty estimation in deep learning models is an important and practical problem for robotics. This paper shows how to tailor Laplace Approximation to 3D object detection task by introducing several new techniques. As a result, the paper suggests an alternative method for episdemic uncertainty estimation over MC-dropout. The findings overall reflect the current state-of-the-art in Bayesian Deep Learning community, and can be a useful reference for their applications in 3D object detection tasks. There are also no notable grammar and spelling mistakes, and limitations are addressed in detail at the end of the paper. Code is also attached, and the video in the supplementary material helps in acquring the main points of the paper.

However, I find that the paper requires significant revision in improving the clarity. Experiments require also significantly more evidence to strengthen the contributions of the paper. Concretely:

1. Line 26-31/56-59: The claim that Laplace Approximation is limited to classification task requires some revision. Laplace Approximation is merely an inference tool to obtain the posterior distribution, and many of the existing works are not limited to classification. Some examples are given below, and I also note that many of the cited paper in that paragraph contains experiments on regression tasks, e.g.,
[1] learning multiplicative interactions with bayesian neural networks for visual-inertial odometry, 2020.
[2] sparse bayesian deep learning for dynamic system identification, 2020.
[3] trust your robots! predictive uncertainty estimation of neural networks with sparse gaussian processes, 2021.
In my view, it would work better to argue and explain specific problems of directly applying laplace approximation to existing 3D object detection architectures, which will lead to three method contributions of this paper, instead of these lines.
On the same note, the argument on output dimenision can be misleading -- there are alternative formulation to get the dimensions to scale with weight dimensions.

2. Line 90-126: some of these basic information can go to appendix, and rather use the space to describe the contributions of the paper. In my view, many details about the contribution is missing in later part of the text, whereas this basic information is rather explained in too much detail. Equally important, stating the challenges in line 123-126 without explanations on why, feels difficult to follow.

3. Algorithm 1: there are so many unintroduced variables. I could not find what these symbols mean in the text as well. I guess that it means some prior parameters which is specified by the user. Howeover, having to guess makes it hard to read the paper. Moreover, the last equation  to compute the fisher, is not the formulation for diagonal approximation right? Computationally, that equation cannot be done due to its size.

4. section 3.1.1/section 3.1.3: As a reader, I find these sections difficult to parse in depth, which is very unfortuante given that these texts are the main contributions of this paper. I do not get the following points:
(a) why do you need to compute fisher as in equation 10, which poses the problem with outer dimensions. Just like cited laplace approximation methods, posterior distribution can be inferred in weight space, and predictive distribution can also be computed through marginalization of the posterior distribution
(b) there exists libraries for laplace approximation and curvature computations. why do you need algorithm 1? "because 3D object detection network is originally determinstic" --> but all the pretrained networks are deterministic, if one does not consider the posterior distribution.
(c) for prior weight determination, line 180, what do you mean by infinitely large value? why 30 percent quantile for each layer? (empirical rule of thumb is not convincing that it will work always). Figure 3 contains lagend that are not explained.
(d) comparion with naive approach also do not reflect the state-of-the-art. How about using bayesian optimization or variational inference like existing literature? e.g., [4] Bayesian optimization meets laplace approximation for robotic introspection, 2020. [5] Sparse Uncertainty Representation in Deep Learning with Inducing Weights, 2021.

5. Regarding experiments, deep ensembles must be added as additional baseline. Comparison only in KITTI is limited -- how about citiscape? This will show that the trend will follow also in other data-sets. Monte carlo run of 10 is limited, and I suggest to try 20-30.

6. Run-time comparison is also needed. The authors state that "The real-time performance can be improved if it parallels the M Monte-Carlo computations with batched input [32]". Howeover, I cannot find the empirical evidence that real-time performance can be reached, or the stack is more efficient. It is highly recommended to implement such module, and report the run-time of all these systems, in order to show that contributions presented here is in principle, applicable to robotics.

**Summary Of Recommendation:**

My recommendation is a weak reject. I appreciated the contribution to adapt existing techniques from machine learning literature to practical robotic vision tasks. Reported problems and adjustments made to the pipeline were interesting (though I was not convinced apart from separation and aggregation of uncertainty for 3D object detection).

However, I find that the presentation of the paper needs significant improvement, in order to comprehend the materials. Experiments require more work in terms of adding more compelling baselines, and evaluating run-time for examining its applicability to robotics. From methdological point of view, the prior weight determination part requries more motivations on design choices, and the fisher estimation using output samples with pre-specified distribution requires more clarifications -- all these points with respect to existing methods.

----------------------------- Update after rebuttal --------------------------

After the rebuttal, I have increased my recommendation to the weak accept. This was due to addition of another dataset for evaluation, and certain clarifications made in the revised paper, e.g., new figures that are helpful to pass the insights, clarifying the main contributions, etc.

Maybe, though, further improving the writing could help. Specially, certain claims also need to be thoroughly checked, like output dimensions posing difficulties (which can be mitigated by working on a weight space formulation with certain approximations like KFAC), or clearer discussion about empirical fisher Vs gradient based ones, etc.

---

> ### Author Response · Authors · 2022-08-22
> **Response to Reviewer-S5g8 (updated)**
>
>
> The authors would like to thank these valuable comments from Reviewer-S5g8. Following contains the response to the comments. Three pdf files are included to clarify these points:
>
> - revised paper (**updated in Aug26**)
> - appendix file (**updated in Aug26**)
> - response file (**updated in Aug26**)
>
> The updated responses are marked with **updated in Aug26**.
>
> ## Regarding to "Line26-31/56-59...with weight dimensions"
>
> Sorry for this misleading claim. It is true that the Laplace approximation can be applied to both classification and regression tasks, just as listed in [1][2][3]. The challenges in applying Laplace approximation in 3D object detection include:
>
>     - The high dimensional output space makes the exact calculation with [Martens2020JMLR] of Fisher intractable.
>
>     - The high dimensional parameter space makes the exact fisher calculation within the weight space intractable. We have to use Monte-Carlo approaches to compute the inner expectation over predictive distribution.
>
>     - For RCNN-based detectors, like PointRCNN, the ranking results in proposal selection might mismatch among different Monte-Carlo samples, which causes the problem that the moment estimation process cannot estimate the true moments in the final prediction distribution of 3D bounding boxes. Therefore, we propose an Uncertainty Separation and aggregation pipeline for the Bayesian inference in two-stage 3D detectors.
>
> [Martens2020JMLR]J. Martens. New insights and perspectives on the natural gradient method. Journal of Machine
> 389 Learning Research, 21:1–76, 2020.
>
> ## Regarding "Line 90-126...difficult to follow."
>
> **updated in Aug26**
>
> Thanks for the comments. We have move the detailed derivations to the appendix. Hope the methodology section in the revised paper can improve the readability.
>
> ## Regarding "Algorithm 1...to its size."
>
> **updated in Aug26**
>
> Thanks for the comments. We have removed this algorithm from the main text in the revised paper.
>
> ## Regarding "(a) why do you need to compute fisher as in equation 10, which poses the problem with outer dimensions."
>
> Equation 10 is not the method to calculate Fisher in this paper. Here we write this equation to analyze why it is inapplicable in the task of 3D object detection. As the reviewer comments, it "poses the problem with outer dimensions".
> In the revised paper, we clarify this point by describing the diagonal standard Fisher and empirical Fisher in Section3.3 in the revised paper.
>
> ## Regarding "(b) there exists libraries for laplace approximation and curvature computations. why do you need algorithm 1?"
>
> **updated in Aug26**
>
> Thanks for the comments. We have removed this algorithm from the main text in the revised paper.
>
> It is true that there exists libraries for Laplace approximation and curvature computations, like backpack, asdfghjkl, Laplace Redux. However, these packages focus on small end-to-end neural networks, seldomly applied to large networks with high dimensional output. In addition, since there exist special operations like sparse 3D convolution and NMS, existing libraries cannot be directly used to apply Laplace approximation in 3D object detectors.
>
> It is not because "3D object detection network is originally deterministic". We mentioned this problem is to complete the description of Algorithm1, and explain those prior parameters. It is not specific challenge or features in evaluating Fisher for 3D object detectors.
>
> ## Regarding "(c) for prior weight determination, line 180, what do you mean by infinitely large value? why 30 percent quantile for each layer? "
>
> **updated in Aug26**
>
> Thanks for the comments. We have removed this section from the main text in the revised paper.
>
> The parameters, whose diagonal Fisher values are smaller than $t$ are considered their Fisher estimation values are inaccurate. They will be given an infinitely large value in $\Sigma_0^{-1}$. It eliminates the effects of inaccurate Fisher elements and will result in zero values in the final covariance matrix $\Sigma$.
> We have updated the description in the Section3 of the revised appendix.
>
> ## Regarding "Figure 3 contains lagend that are not explained"
>
> Sorry for the legend problem. They have been fixed in FigureA2 of the revised appendix.
>
> ## Regarding "Regarding experiments...in other data-sets"
>
> We also conduct experiments on the NuScenes dataset. The result is in the Section6 of the revised appendix.
>
> ## Regarding "Monte carlo run of 10 is limited"
>
> **updated in Aug26**
>
> We also conduct experiments with the number of Monte Carlo samples as 20. The results are reported in the Table-R2-1 in the response file.
> The Figure A5 in the appendix file also shows the NLL performance under various number of Monte-Carlo samples. The same conclusion still holds in a wide range of number of Monte Carlo samples.

---

> ### Author Response · Authors · 2022-08-22
> **Response to Reviewer-S5g8 (updated)**
>
>
> ## Regarding "Run-time comparison...applicable to robotics"
>
> We plot the run-time performance and memory consumption in the Section 10 of the revised appendix without parallelling the Monte Carlo computations. The speed can be improved if we parallel the calculation of different Monte-Carlo inferences, which requires us to re-implement most of the code implementation. Since the run-time performance is not the major contribution we want to claim in this paper, we will temporarily leave this real-time parallel implementation in the latter part of this discussion period.
>
> ## Regarding "Line 187-193...with empirical data?"
>
> **updated in Aug26**
>
> Point1 evaluates the predictive distribution of the Laplace-approximation-based probabilistic 3d object detection with ECE and NLL metrics in the revised paper.
> The original point3 is to better understand the estimated predictive distributions, and compare them with the generative model proposed in [Wang2020IROS]. The generative model [Wang2020IROS] characterizes the generative process of key-object points within 3D bounding boxes.
> In the revised paper, we clarify this point at the beginning of Section4 and Section4.3
>
> [Wang2020IROS] Z. Wang, D. Feng, Y. Zhou, L. Rosenbaum, F. Timm, K. Dietmayer, M. Tomizuka, and W. Zhan. Inferring spatial uncertainty in object detection. In 2020 IEEE/RSJ International Conference on Intelligent Robots and Systems (IROS), pages 5792–5799, 2020.
>
> ## Regarding "Error bars needed...table 2 and figure 5."
>
> Thanks for the comments.
> Figure 5 has been updated with error bars in the Figure4 of the revised paper.
> We report the mean values of 5 random seeds in the Table 3 of the revised paper, and present the std values in the Table A2 of the revised appendix file.

---

> ### Author Response · Authors · 2022-08-22
> **Response pdf file (updated)**
>
> The attachment contains the response pdf file **updated in Aug26**.

---

### Official Review · Reviewer_dWMY · 2022-08-01

**Originality:** Good
**Technical Quality:** Good
**Clarity Of Presentation:** Poor
**Impact:** 3

**Recommendation:**

Weak Reject: I recommend rejecting the paper, but will not argue for my recommendation if the majority of other reviewers have a different opinion.

**Summary:**

This paper describes an approach for estimating the epistemic uncertainty of DNN predictions for 3D object detection.  The main technical contribution is to propose a strategy for handling two properties of 3D object detectors that make it difficult to apply standard (Laplace approximation-based) uncertainty quantification methods: (i) the high dimensionality of the outputs (which makes calculating the required Hessian or Fisher Information Matrix prohibitively costly), and (ii) the fact that many of these models include nondifferentiable selection and ranking operations (which means that the overall predictors are not end-to-end differentiable, as required in order to evaluate the Hessian used in the Laplace approximation).  The paper proposes an approach to address these challenges using (i) Monte Carlo approximation and (ii) an uncertainty separation and aggregation method that splits over the nondifferentiable operations in two-stage predictors, respectively.


**Issues:**

The clarity of the writing needs to be improved throughout, but especially in the presentation of the main technical results.  In general, “natural language” is a very poor medium for communicating algorithms or mathematics (precisely because it *is* so vague).  I would strongly encourage the authors to replace these narrative descriptions with precise mathematical statements.


**Quality Of The Limitations Section:**

Limitations are addressed clearly

**Reviewer Expertise:**

3: The reviewer is fairly confident that the evaluation is correct

**Robotics Focus:**

Sufficient demonstration on hardware

**Strengths And Weaknesses:**

**Strengths:**  The paper addresses a problem that is clearly important (as it has safety implications for autonomous robots, among other applications) and proposes what appears to be a technically well-founded and clever approach to handling the nondifferentiability of two-stage 3D detectors by showing how to decompose the calculation of the required predictive distribution in a way that avoids the need to propagate derivatives through the (discrete) selection.  The experimental results also seem to show that the proposed method is practically effective, especially in comparison with prior alternatives.

**Weaknesses:**  The paper’s most prominent weakness is that (in my opinion) it is very difficult to follow, primarily due to a lack of clarity and precision in the technical development.

For example, the description of how to approximate the Fisher Information Matrix (presented in Section 3.1.1) is given primarily in the form of a (in my opinion quite vague) “natural language” description in a paragraph of text in lines 127 - 143; the only description approaching a precise mathematical or algorithmic formulation is Algorithm 1, which contains a joint distribution (in the calculation of the gradient) that i don’t think is actually defined in the paper.

The description of the uncertainty aggregation and separation approach is similarly opaque; once again this primarily takes the form of a long (similarly imprecise) “natural language” description in a pair of paragraphs in lines 144-170, which in turn reference a pictograph (Fig. 2) that I find to be basically incomprehensible.


I highlight these two examples because these portions of the paper are supposed to be explaining its *two principal contributions*, yet they are so inscrutable that I am very skeptical a typical reader would actually be able to implement the approach they are meant to describe.   (Large portions of the experimental results section are similarly imprecise.)

Unfortunately it is difficult for me to meaningfully comment on other aspects of the work because (due to this lack of clarity in the writing) I am not confident that I understand the proposed method well enough to offer an informed opinion.


**Summary Of Recommendation:**

I recommend rejection on the basis that the paper as currently written does not effectively communicate the technical information that is supposed to be its principal contribution.

---

### Meta-Review · Area_Chair_ivgB · 2022-08-09

**Recommendation:** Accept (Poster)
**Confidence:** 5

**Metareview:**

All reveiwers appreciate the relevance of the uncertainty estimation problem to the deep learning domain and its applicability in robotics. In particular, the proposed approach for 3D object detection is interetsing and extends current LA based techniques to this important problem. However, there are also important concerns regarding the clarity of the descripton and the experimental evaluation. For the former, a more mathematically concise formulation is required with a better definition of symbols and less descriptive "natural" language. For the latter, experimental comarisons to state-of-the-art methods such as deep ensembles or K-FAC are necessary.

Post-rebuttal:
The authors have done a good job addressing the points raised by the reviewers. They have shown additional experimental results in the revised version of the paper, which has improved significantly. As the main open point, it remains the question why the theoretically not fully motivated approach of the empirical Fisher often performs better that the theoretically more sound method using the standard Fisher. In the rebuttal, the authors claim that this is due to the diagonal approximation used in the standard Fisher, but this needs to be verified at least empirically. Still, as it is, the paper is a worthwile contribution to the field of uncertainty-aware robotic perception and an interesting uncertainty-based extension to the problem of 3D object detection.

**Best Paper Nomination:**

No

---

> ### Author Response · Authors · 2022-08-27
> **Response to AC and reviewers**
>
> Thanks for the efforts and valuable comments from our reviewers and AC.
> In this discussion period, we responded to all our reviewers' comments and provided additional experiment results and analyses to improve the paper.
> To improve the description clarity, we revise the paper with concise math formulations.
> To strengthen the experiment evaluation, we add comparisons with the deep-ensembles baseline on the real-world KITTI dataset.
> In addition, we also include the experiment results and analysis required by our reviewers in the appendix to improve the quality of this work.
> It is our hope that the proposed Laplace approximation method, evaluation protocol, baselines, and code can boost the community to explore probabilistic object detection.
>
> The attachment contains our revised paper and appendix file.
> For easy reading, the major changes have been highlighted in blue in the revised paper.

---

> ### Author Response · Authors · 2022-09-13
> **Thank you all**
>
> Dear Chairs and reviewers,
>
> The authors would like to thank Chairs and reviewers for their efforts and insightful comments!
>
> Thanks & Best Regards,
>
> Authors of Paper 193